# Association of Functional Characteristics and Physiotherapy with COVID-19 Mortality in Intensive Care Unit in Inpatients with Cardiovascular Diseases

**DOI:** 10.3390/medicina58060823

**Published:** 2022-06-18

**Authors:** Chiara Andrade Silva, Agnaldo José Lopes, Jannis Papathanasiou, Luis Felipe Fonseca Reis, Arthur Sá Ferreira

**Affiliations:** 1Postgraduate Program of Rehabilitation Sciences, Centro Universitário Augusto Motta/UNISUAM, Rio de Janeiro 20080-003, Brazil; chiara.andrade@yahoo.com.br (C.A.S.); alopes@souunisuam.com.br (A.J.L.); luisfelipefreis@souunisuam.com.br (L.F.F.R.); 2Department of Medical Imaging, Allergology & Physiotherapy, Faculty of Dental Medicine, Medical University of Plovdiv, 4002 Plovdiv, Bulgaria; giannipap@yahoo.co.uk; 3Department of Kinesitherapy, Faculty of Public Health “Prof. Dr. Tzecomir Vodenicharov, DSc.”, Medical University of Sofia, 1431 Sofia, Bulgaria

**Keywords:** cardiovascular care, cardiovascular hospitalization, physiotherapy, physical therapy, inpatient rehabilitation, SARS-CoV-2

## Abstract

*Background and Objectives:* To estimate the association between admission functional outcomes and exposure to physiotherapy interventions with mortality rate in intensive care unit (ICU) inpatients with cardiovascular diseases and new coronavirus disease (COVID-19). *Materials and Methods*: Retrospective cohort including 100 ICU inpatients (mean (standard deviation), age 75 (16) years) split into COVID-19+ or COVID-19−. The association of in-ICU death with admission functional outcomes and physiotherapy interventions was investigated using univariable and multivariable regression models. *Results:* In total, 42 (42%) patients tested positive for COVID-19. In-ICU mortality rate was 37%, being higher for the COVID-19+ group (odds ratio, OR (95% CI): 3.15 (1.37–7.47), *p* = 0.008). In-ICU death was associated with lower admission ICU Mobility Scale score (0.81 (0.71–0.91), *p* = 0.001). Restricted mobility (24.90 (6.77–161.94), *p* < 0.001) and passive kinesiotherapy (30.67 (9.49–139.52), *p* < 0.001) were associated with in-ICU death, whereas active kinesiotherapy (0.13 (0.05–0.32), *p* < 0.001), standing (0.12 (0.05–0.30), *p* < 0.001), or walking (0.10 (0.03–0.27), *p* < 0.001) were associated with in-ICU discharge. *Conclusions:* In-ICU mortality was higher for inpatients with cardiovascular diseases who had COVID-19+, were exposed to invasive mechanical ventilation, or presented with low admission mobility scores. Restricted mobility or passive kinesiotherapy were associated with in-ICU death, whereas active mobilizations (kinesiotherapy, standing, or walking) were associated with in-ICU discharge in this population.

## 1. Introduction

Cardiovascular diseases (CVDs) are the leading cause of death globally, with estimated 31% mortality, representing about 17.9 million deaths every year [1]. In Brazil, CVDs were responsible for 267,635 (29.5% of all causes) deaths in 1990 and 424,058 (31.2% of all causes) deaths in 2015 [2]. The Brazilian population experienced the global trend of a reduction in the risk of death from CVD compared to the 1990s [3]. With the recent flattening of the rate of decline in CVD mortality in Brazil, the research on CVD and its functional repercussion gained additional momentum [4]. The populational aging and therapeutic advances in the treatment of CVD lead to longer survival and consequent increase in the prevalence of CVD, as well as frequent hospitalizations [5]. In December 2019, an outbreak of pneumonia due to the severe acute respiratory syndrome coronavirus 2 (SARS-CoV-2) was detected in the city of Wuhan, China, which spread globally to a pandemic by 11 March 2020. A higher incidence of SARS-CoV-2 infection is being observed in patients with CVD, as well as manifestations of more severe symptoms, increased hospital length of stay and increased risk of death among patients hospitalized with the new coronavirus disease (COVID-19) [6,7,8,9].

The hazards of hospitalization, particularly for older adults in the intensive care units (ICU), are a longstanding issue that ultimately favors a decline in musculoskeletal function and functional capacity [10]. Higher overall muscle strength evaluated at ICU admission and lower total length of stay are predictors of functional improvements at both ICU and ward settings [11]. Muscle strength is an intrinsic factor related to the postural balance and risk of falling [12] and is also associated [13] with the changes in mobility in older adults after acute hospitalization. The functional status of patients hospitalized with CVD and COVID-19 remains largely unknown and warrants urgent investigation [14]. Patients with COVID-19 can improve their mobility at hospital discharge and have a higher probability of discharging home with increased frequency and longer mean duration of physiotherapy visits [15]. Nonetheless, whether functional characteristics at ICU admission and physiotherapy interventions can affect the risk of in-ICU death remains unclear in this population. Therefore, this retrospective observational study investigated whether admission functional outcomes and exposure to physiotherapy interventions are associated with in-ICU mortality rate in older adults with cardiovascular diseases and COVID-19.

## 2. Materials and Methods

### 2.1. Study Design and Reporting

Retrospective, single-center study. Data were obtained by the principal investigator through information previously contained in electronic medical records, examination reports, and notes of the health professional staff involved in the care of the patients. This study is reported following the REporting of studies Conducted using Observational Routinely-collected health Data (RECORD) statement [16]. Minimum sample size was determined to ensure precision of the estimate of overall risk using logistic regression models [17]. A minimum of 96 participants is required to ensure a margin of error ≤ 0.1 for a true outcome proportion equal to 0.5.

### 2.2. Setting and Participants

This study retrospectively analyzed all data from February to November 2020 collected from patients consecutively hospitalized at the ICU of a primary-to-tertiary private hospital located in Curitiba, Paraná, Brazil.

ICU admission criteria comprised at least of the following conditions: hemodynamic (e.g., symptomatic hypotension, hypertension with target-organ damage; hypovolemia, perfusion impairment of any etiology); neurological (e.g., lowering of consciousness level, intoxication, ischemic or hemorrhagic stroke, convulsive crisis); or respiratory (e.g., hypoxemia, respiratory failure, pneumothorax, pulmonary edema, dyspnea). Other causes for ICU admission included postoperative period of high-risk patients, postoperative period of large surgeries, postoperative cardiac surgery, postoperative neurological surgery, postoperative endovascular surgery, need for clinical monitoring, sepsis, septic shock, cardiac arrhythmia, vascular diseases of the heart, or acute renal dysfunction.

Patients who had a primary diagnosis of CVD after a complete clinical exam and laboratory testing including laboratory blood tests, electrocardiogram, blood pressure, and/or echocardiography as prescribed, admission assessment by a physiotherapist, and tested for SARS-CoV-2 infection at admission were included. ICU admission was defined as an admission to the hospital’s ICU for >12 h. Re-admissions of patients to the ICU within the study period were excluded from the analysis.

### 2.3. Clinical Measurements

All admission data were collected within <24 h of ICU hospitalization at the discretion of the medical staff and covered the required time for swab analysis. Data were collected retrospectively from electronic medical recordings regarding demographics, vital signs, laboratory, gasometry, presence of CVD and comorbidities, and drugs in continuous use. Date of hospital admission and discharge from the ICU or death were collected for computing the total length of ICU stay. The sample was divided into groups COVID-19+ or COVID-19− based on the ICU admission test, after a nasal and/or nasopharyngeal swab for SARS-CoV-2 by polymerase chain reaction method.

### 2.4. Functional Measurements at ICU Admission

Overall muscle strength was assessed by the Medical Research Council (MRC) scale, which uses a 6-point scale of 6 muscle groups bilaterally. Representative scores comprised the sum of points observed for each muscle group bilaterally, ranging from 0 (no muscle activity) to 60 (maximal muscle strength) [18].

Mobility was assessed by the ICU Mobility Scale (IMS). The score varies between 0 expressing low mobility (patient who only performs passive exercises in bed) and 10 expressing high mobility (patient who presents independent walking, without assistance) [19].

### 2.5. Physiotherapy Interventions

All patients were exposed to physiotherapy interventions based on the hospital standards of usual care developed according to international and national recommendations [20,21,22,23,24,25,26,27,28,29], which are briefly described here. Full descriptions of the hospital standards for usual care are available in Brazilian Portuguese upon request to the authors. In brief, the following routine was applied daily. First, a multidisciplinary rehabilitation meeting was held daily to link the plans between the professionals of each patient. In sequence, an initial bedside assessment of vital signs, pain intensity, ventilatory pattern, level of consciousness, and dosage of drug infusion in use was conducted to decide upon the removal of patients from the bed and/or which physiotherapy intervention to perform. In general, physiotherapy was performed 2 to 3 times daily (one service at each staff shift), involving respiratory and mobility interventions. Ventilatory support adjustments were made according to clinical status and laboratory tests; asynchronous adjustments or parameter changes for weaning were personalized. Exposure to each routine physiotherapy intervention was defined as using a given therapeutic resource at any time during the total length of stay in the ICU, thus registered as dichotomous variables (‘yes’ = 1; ‘no’ = 0).

Ventilatory support was characterized using non-invasive mechanical ventilation, through an orofacial or facial interface connected to the mechanical ventilator in one or two pressure levels ventilation modes, or invasive ventilatory support (orotracheal or tracheal prosthesis in controlled ventilatory modes, controlled assistance, and/or spontaneous). Patients diagnosed with acute respiratory distress syndrome used protective strategy ventilatory parameters, which may require alveolar recruitment through the prone position or recruitment through the gradual increase of positive end-expiratory pressure (PEEP) up to 35 cmH_2_O and subsequent titration of ideal PEEP, provided they were clinically stable. When they needed oxygen therapy, it was performed using a low-flow system (nasal catheter, face mask with reservoir, tracheostomy mask). Spontaneous prone was also used for at least 1 h. In the supine position [30], the head was elevated between 30° and 45°. In the prone position, the head was elevated between 10° and 20°.

Mobility activities were categorized as complete bed restriction; passive kinesiotherapy (the physiotherapists passively mobilized the wrist, elbow, shoulder, hip, knee and ankle joints, stretching and positioning the individual to bed); active kinesiotherapy (active free, active resisted or assisted active mobilization of the wrist, elbow, shoulder, hip, knee and ankle joints, dynamic or static global stretches, trunk control work); assisted or active sitting out of bed; standing; and walking.

### 2.6. Study Outcomes

The primary outcome was in-ICU mortality as well as admission functional assessments of MRC and IMS scores. In-ICU mortality was calculated from the admission date and confirmed using electronic medical records.

### 2.7. Data Access and Cleaning Methods

No missing data occurred for exposures, in-ICU mortality, or admission IMS scores. Data were missing for the admission assessment of MRC scores in 18/100 participants due to sedation.

### 2.8. Statistical Analysis

Statistical analysis was performed in jamovi v. 1.8.1.0 and R project version 4.0.4 with packages after importing the electronic spreadsheet. Missing data in admission measurements were reported and assumed to be missing completely at random and univariate mean imputation was performed. Evidence of statistical significance was considered at *p* < 0.05 (two-tailed).

Descriptive summaries were reported as mean (standard deviation (SD)) for continuous variables or absolute and relative frequencies (%) for categorical ones. Admission demographic data were compared between COVID-19+ versus COVID-19− groups using the linear model analysis of variance or Pearson’s Chi-squared test for continuous and dichotomous variables, respectively.

Univariable logistic regression analysis was performed to examine the association (odds ratio (OR) with 95% confidence interval (95% CI)) of exposure to physiotherapy intervention (ventilatory support and mobility) with group (COVID-19+ vs. COVID-19−) and in-ICU mortality. Multivariable logistic regression model was fitted to determine independent factors associated with in-ICU mortality; all factors related to exposure to physical therapy were force-entered as a full model. Model fit was evaluated by Akaike information criterion (AIC) and C-statistic.

## 3. Results

### 3.1. Sample Characteristics

Table 1 shows the demographic and clinical data of the studied sample. A total of 108 inpatients with CVD were retrieved; of these, 8 records were excluded as they were re-hospitalizations, resulting in 100 participants for analysis. Underlying diagnoses that required hospitalization in the ICU were likely multiple per patient and comprised pulmonary sepsis or pneumonia (43/100); decompensated heart failure (13/100); acute kidney failure (13/100); dyspnea (9/100); exacerbation of chronic pulmonary obstructive disease (7/100); severe acute respiratory syndrome (6/100); stroke (5/100); pulmonary embolism (5/100); acute respiratory failure, acute myocardial infarction, femur fracture, chest pain, urinary tract infection (each 3/11); abdominal pain, minimally conscious state, electrophysiological study, sepsis (each 2/100); and exploratory laparotomy, lower gastrointestinal bleeding, diabetic ketoacidosis, acute pancreatitis, syncope, soft tissue sepsis, post-surgery shoulder arthroplasty, post-surgery atrial ablation, diverticulitis, acute tachycardia, post-surgery spinal cord stimulator implantation, acute cholecystitis, atrial flutter, atrioventricular block, deep venous thrombosis, post-surgery pericardial drainage, hypoxia, pleural effusion, abdominal sepsis, bronchospasm, atrial fibrillation with rapid ventricular response, and anemia (each 1/100).

Overall, the sample was composed of older adults (75 (16) years), balanced between sexes (female/male 49/51), and with most participants (40/100) classified as overweight (body mass index = 27.1 (5.1) kg/m^2^). The most common underlying CVD was hypertension (91/100), followed by a history of cerebrovascular disease (22/100), coronary artery disease (21/100), heart failure (16/100), and atrial fibrillation. Overall length of ICU stay was 9.7 (15.5) days, with patients with COVID-19+ showing longer ICU stay (14.5 (21.7) vs. 6.2 (6.8), *p* = 0.007). Between-groups comparisons show that patients with COVID-19+ showed higher admission MRC scores (48.3 (7.9) vs. 43.8 (10.4), *p* = 0.021) and IMS scores (5.5 (4.0) vs. 3.9 (3.8), *p* = 0.039). They were also younger (68 (16) vs. 80 (13) years, *p* < 0.001) and presented at admission with lower leukocytes count (9565 (5473) vs. 13,456 (6444) count/mcL, *p* = 0.002), lower partial pressure of carbon dioxide (PCO_2_) (31.7 (6.5) vs. 37.7 (8.5), *p* < 0.001), and lower bicarbonate (21.9 (4.5) vs. 23.8 (4.9) mEq/L, *p* = 0.044). Sedation at admission was more frequent in the COVID-19+ group (52.4% vs. 39.0%, *p* = 0.020). No statistical evidence of difference was observed between groups for severity of disease based on acute physiology and chronic health evaluation (APACHE II: 30.3 (4.8) vs. 30.2 (4.9), *p* = 0.917).

### 3.2. Outcomes and Factors Associated with COVID-19+ Test Result

As related to ventilatory support (Table 2), patients with COVID-19+ were more likely exposed to invasive mechanical ventilation in either supine (2.92 (1.29–6.81), *p* = 0.011) or prone position (22.80 (4.19–425.35), *p* = 0.003), to alveolar recruitment (19.04 (4.97–125.99), *p* < 0.001), awake prone (5.60 (1.27–39.05), *p* = 0.038), or length of stay (1.07 (1.02–1.13), *p* = 0.008), but not to non-invasive mechanical ventilation or oxygen therapy. Multivariable logistic regression analysis showed a good linear fit (C-statistic = 0.783), with COVID-19 remaining independently associated with exposure to alveolar recruitment (22.34 (3.56–224.91), *p* = 0.002) or awake prone (13.41 (1.62–228.22), *p* = 0.032).

As related to mobility during ICU stay (Table 2), no statistical evidence of significant differences in odds was observed between groups COVID-19+ and COVID-19− regarding exposure to restricted mobility, passive, or active kinesiotherapy, and standing or walking activities. Multivariable logistic regression analysis showed a good linear fit (C-statistic = 0.694), with no statistical evidence of differences between groups except for length of stay remained (1.07 (1.01–1.14), *p* = 0.029).

### 3.3. Outcomes and Factors Associated with in-ICU Mortality

Table 3 presents the association of in-ICU death and admission functional measurements with exposure to the physiotherapy interventions. In-ICU mortality rate was 37%, being higher for the COVID-19+ group (3.15 (1.37–7.47), *p* = 0.008). As related to ventilatory support, in-ICU mortality was more likely in patients exposed to invasive mechanical ventilation in either supine (22.71 (8.28–70.76), *p* < 0.001) or prone position (4.74 (1.42–18.74), *p* = 0.016), oxygen therapy (8.75 (2.34–57.11), *p* = 0.005) or alveolar recruitment (10.06 (3.25–38,35), *p* < 0.001). Also, in-ICU death was associated with longer length of stay (1.04 (1.01–1.09), *p* = 0.048)), lower admission IMS score (0.81 (0.71–0.91), *p* = 0.001) but not MRC score (*p* = 0.055). Multivariable logistic regression analysis showed an excellent linear fit (C-statistic = 0.920), with in-ICU death still independently associated with COVID-19+ (5.51 (1.25–28.43, *p* = 0.029), exposure to invasive mechanical ventilation (14.81 (2.97–97.70), *p* = 0.002) and admission IMS score (0.79 (0.63–0.96), *p* = 0.023).

As related to mobility during ICU stay, two patterns emerged: restricted mobility (24.90 (6.77–161.94), *p* < 0.001), passive kinesiotherapy (30.67 (9.49–139.52), *p* < 0.001) and longer LOS (1.04 (1.01–1.09), *p* = 0.048) were associated with in-ICU death, whereas active kinesiotherapy (0.13 (0.05–0.32), *p* < 0.001), standing (0.12 (0.05–0.30), *p* < 0.001), or walking (0.10 (0.03–0.27), *p* < 0.001) were associated with in-ICU discharge. Multivariable logistic regression analysis showed an excellent linear fit (C-statistic = 0.947), with in-ICU death still independently associated with COVID-19+ (15.44 (2.80–140.82), *p* = 0.005), restricted mobility (10.49 (1.74–95.85), *p* = 0.017), and passive kinesiotherapy (17.66 (2.32–326.56), *p* = 0.017).

## 4. Discussion

The main findings suggest that in-ICU mortality is higher for inpatients with CVD who had a COVID-19+ test result, were exposed to invasive mechanical ventilation, or presented with low admission IMS scores, whereas restricted mobility or passive kinesiotherapy were associated with in-ICU death, active mobilizations (kinesiotherapy, standing, or walking) were associated with in-ICU discharge. Our findings contribute to the global discussion on the acute management of patients with COVID-19 by providing insights about the planning physiotherapy interventions aimed at reducing in-ICU fatality among patients with CVD.

Demographic and clinical data from our sample in Brazil corroborate previous studies in other countries on the risk factors for hospitalization and mortality in patients with CVD and COVID-19: mainly older age, overweight, low lymphocyte count, and pre-existing comorbidities, among others [6,7,8,9,31,32]. In Brazil, the rate of population aging helps to explain the predominance of non-communicable chronic diseases as the main causes of hospitalization and death in older individuals [33]. Overall length of stay of our sample was similar to other studies in patients with COVID-19, ranging from <1 week to 2 months [34]. A retrospective study including 88 older adults hospitalized for COVID-19 conducted in an ICU Brazil reported hypertension as the most common comorbidity and a median length of ICU stay of 23 days (4–38) [35]. The link between pre-existing CVDs with worse outcomes and increased risk of death in patients with COVID-19 is also corroborated by our findings [36]. Altogether, these results suggest an external validity of our findings while highlighting the role of demographic characteristics and COVID-19 diagnosis associated with in-ICU death in this population.

Clinical algorithms [37] and consensus [38] for the respiratory management of patients with COVID-19 are emerging. Our findings contribute to the further development of those algorithms as they suggest that inpatients with CVD and COVID-19+ were more likely exposed to ventilatory support techniques, particularly alveolar recruitment (concomitant with invasive ventilatory support) and awake prone. Nonetheless, the role of early mobilization in patients with COVID-19 is already acknowledged [38,39], but algorithms that include mobility interventions for this population are still lacking. While the similar exposure to all mobility interventions reinforces the longstanding overall need for early mobilization in hospitalized patients [40], the higher exposure to passive kinesiotherapy in patients with COVID-19+ may represent a proxy for disease severity in this group.

In-ICU mortality was higher for inpatients with CVD who had a COVID-19+ test result, were exposed to invasive mechanical ventilation, or presented with a lower admission IMS score. Altogether, these characteristics can be understood as a proxy for disease severity. Interestingly, exposure to physiotherapy intervention showed two distinct effects on in-ICU mortality rate, whereas restricted mobility or passive kinesiotherapy were associated with in-ICU death, active mobilizations (kinesiotherapy, standing, or walking) were associated with in-ICU discharge. This finding corroborates previous studies showing improved mobility at hospital discharge and a higher probability of discharging home with increased frequency and longer mean duration of physical therapy visits in patients with COVID-19 admitted to acute care hospitals [15]. Considering the interventions investigated herein listed in an ‘ordered’ fashion indicating a patient’s recovery—i.e., progressing from restricted mobility to passive kinesiotherapy, to active kinesiotherapy, and so on—it can be argued that crossing the ‘passive-to-active kinesiotherapy’ threshold can be a major factor to change the clinical course and possibly the outcome. Further studies are necessary to investigate whether different sequences of exposures to physiotherapy intervention are associated with in-ICU mortality and, if so, what sequential path is more likely associated with in-ICU discharge.

This study has some limitations. Due to the retrospective design, there were missing data for participants regarding admission assessment of functional outcomes. Clinical data at admission were collected within <24 h of ICU hospitalization and hence may differ from pre-admission status that required hospitalization. Moreover, physiotherapy interventions were delivered as per the rehabilitation team’s clinical decision-making process. Whereas such lack of control in experimental factors may have influenced the delivered interventions in each group, such pragmatic approach most likely represents ICU routines in Brazil since they are based on national guidelines. Some CIs returned wide ranges, which suggests a large uncertainty about the effect (likely due to small cell counts for some predictors) and that further information is needed. Nonetheless, assessment of goodness-of-fit of the models (Akaike information criterion and C-statistics) suggest acceptable model validity. The current sample is from a single center during the first ‘wave’ (February to November 2020) of cases in Brazil [41] when variants of concern P.1 and B.1.1.7 were the most prevalent ones [42] and may not reflect the nationwide healthcare system, and thus requires further investigation. Finally, we did not explore factors other than clinical characteristics of the groups, ventilatory support, and exposure to physiotherapy interventions that might explain the present findings. Given global reports [43] of shortage of health professionals and lack of personal protective equipment, it is important to emphasize that the physiotherapists were staffed on a 1:10 ratio for ICU beds and had unrestricted access to personal protective equipment during this study period.

## 5. Conclusions

In-ICU mortality is higher for inpatients with CVD who had COVID-19+, were exposed to invasive mechanical ventilation, or presented with low admission mobility scores. The protective effects of routine physiotherapy interventions are highest when patients can perform active rather than passive kinesiotherapy. Patients with COVID-19 who further perform standing and walking activities appear to experience a higher survival effect.

## Figures and Tables

**Table 1 medicina-58-00823-t001:** Summary and comparison of demographic data of inpatients with cardiovascular disease hospitalized with or without COVID-19 (*n* = 100).

	COVID-19−	COVID-19+	Total	*p* Value
	(*n* = 58)	(*n* = 42)	(*n* = 100)	
**Length of ICU stay, days**	6.2 (6.8)	14.5 (21.7)	9.7 (15.5)	0.0071
**Glasgow, score**	13 (3.0)	14 (1.5)	14 (2.5)	0.0561
**APACHE II, score**	30.3 (4.8)	30.2 (4.9)	30.2 (4.8)	0.9171
**Admission functional outcomes**				
*MRC*, *score*	43.8 (10.4)	48.3 (7.9)	45.7 (9.6)	0.0211
*IMS*, *score*	3.9 (3.8)	5.5 (4.0)	4.6 (3.9)	0.0391
**Age, years**	80.4 (13.4)	68.1 (16.2)	75.2 (15.8)	<0.001 ^1^
**Sex, *n***				0.1472
**Female**	32 (55.2%)	17 (40.5%)	49 (49.0%)	
**Male**	26 (44.8%)	25 (59.5%)	51 (51.0%)	
**Body mass, kg**	72.3 (17.4)	81.9 (18.7)	76.3 (18.5)	0.0101
**Body height, m**	1.6 (0.1)	1.7 (0.1)	1.7 (0.1)	0.0061
**Body mass index, kg/m^2^**	26.4 (4.7)	28.2 (5.4)	27.1 (5.1)	0.0791
**Body mass index category, *n* (%)**				0.4512
*Thin*	3 (5.2%)	1 (2.4%)	4 (4.0%)	
*Eutrophic*	19 (32.8%)	11 (26.2%)	30 (30.0%)	
*Overweight*	25 (43.1%)	15 (35.7%)	40 (40.0%)	
*Obesity I*	7 (12.1%)	10 (23.8%)	17 (17.0%)	
*Obesity II*	4 (6.9%)	4 (9.5%)	8 (8.0%)	
*Obesity III*	0 (0.0%)	1 (2.4%)	1 (1.0%)	
**Vital signs**				
*Heart rate*, *beat/min*	84.4 (21.8)	85.5 (17.6)	84.9 (20.0)	0.7941
*Respiratory rate*, *cycle/min*	21.9 (5.5)	21.9 (5.3)	21.9 (5.4)	0.9961
*Systolic pressure*, *mmHg*	137.5 (26.1)	129.5 (25.0)	134.1 (25.8)	0.1271
*Diastolic pressure*, *mmHg*	76.9 (18.8)	75.2 (16.6)	76.2 (17.9)	0.6431
*Pulse pressure*, *mmHg*	60.6 (23.1)	54.3 (17.4)	58.0 (21.0)	0.1391
*Mean pressure*, *mmHg*	97.1 (18.6)	93.3 (18.0)	95.5 (18.3)	0.3111
**Laboratory exams**				
*Sodium*, *mEq/L*	135.8 (6.4)	135.5 (6.4)	135.6 (6.3)	0.8131
*Potassium*, *mEq/L*	4.3 (0.8)	4.2 (0.8)	4.3 (0.8)	0.5531
*Urea*, *mg/L*	69.6 (56.0)	71.5 (69.7)	70.4 (61.8)	0.8821
*Creatinine*, *mg/L*	1.7 (1.8)	1.5 (1.3)	1.6 (1.6)	0.6541
*Lactate*, *mg/L*	1.9 (1.3)	1.6 (0.9)	1.7 (1.1)	0.1891
*Reactive-C protein*, *CP/μL*	75.9 (90.7)	108.8 (96.0)	89.7 (93.9)	0.0841
*Hemoglobin*, *g/dL*	13.0 (2.2)	12.7 (2.3)	12.9 (2.2)	0.5821
*Hematocrit*, *%*	37.6 (6.2)	37.2 (7.2)	37.4 (6.6)	0.7751
*Leukocyte*, *per mcL*	13,456.8 (6443.9)	9564.6 (5472.5)	118,22.1 (6327.6)	0.0021
*Platelets*, *per mcL*	193,869 (80,822)	177,255 (73,991)	186,891 (78,078)	0.2961
*Lymphocytes*, *%*	15.5 (9.3)	15.7 (9.2)	15.6 (9.2)	0.9221
*Neutrophiles*, *%*	78.4 (10.3)	77.4 (11.1)	78.0 (10.6)	0.6661
**Gasometry**				
*pH*	7.4 (0.1)	7.4 (0.1)	7.4 (0.1)	0.0591
*PCO_2_*, *mmHg*	37.7 (8.5)	31.7 (6.5)	35.2 (8.2)	<0.001 ^1^
*Bicarbonate*, *mEq/L*	23.8 (4.9)	21.9 (4.5)	23.0 (4.8)	0.0441
*PaO_2_*, *mmHg*	100.2 (43.6)	89.4 (38.6)	95.7 (41.7)	0.2031
*Base excess*, *mEq/L*	−0.5 (5.3)	−1.5 (5.1)	−0.9 (5.2)	0.3341
*O_2_ saturation*, *%*	95.1 (5.0)	93.9 (6.0)	94.6 (5.4)	0.2891
**Comorbidities, *n* (%)**				
*Hypertension*	55 (94.8%)	36 (85.7%)	91 (91.0%)	0.1162
*Stroke*	15 (25.9%)	7 (16.7%)	22 (22.0%)	0.2732
*Coronary artery disease*	14 (24.1%)	7 (16.7%)	21 (21.0%)	0.3652
*Congestive heart failure*	13 (22.4%)	3 (7.1%)	16 (16.0%)	0.0402
*Atrial fibrillation*	13 (22.4%)	2 (4.8%)	15 (15.0%)	0.0152
**Drugs, *n* (%)**				
*Vasoactive drug*	20 (34.5%)	22 (52.4%)	42 (42.0%)	0.0732
*Sedation*	17 (29.3%)	22 (52.4%)	39 (39.0%)	0.0202

Data shown as mean (SD) or absolute frequency (relative frequency %). ^1^ Linear Model analysis of variance. APACHE: acute physiology and chronic health evaluation; PaO_2_: partial pressure of oxygen. Bold formatting represents grouped variables. Italic formatting represents individual variables within a group.

**Table 2 medicina-58-00823-t002:** Association (odds ratio and 95% confidence intervals) of COVID-19 test (positive vs. negative) and the exposure to the physiotherapy interventions (yes vs. no) estimated using univariable and multivariable logistic regression models (*n* = 100).

Exposures	All Participants	Groups		OR (95% CI)	
		COVID-19−	COVID-19+	Univariable	Multivariable
**Model 1, Ventilatory support**				**AIC = 118, C-statistic = 0.783**	
*Invasive mechanical ventilation*	40 (40%)	17 (43%)	23 (57%)	2.92 (1.29–6.81), *p* = 0.011	0.45 (0.09–1.81), *p* = 0.291
*Invasive mechanical ventilation, in prone*	13 (13%)	1 (8%)	12 (92%)	22.80 (4.19–425.35), *p* = 0.003	5.33 (0.50–134.19), *p* = 0.207
*Noninvasive mechanical ventilation*	13 (13%)	7 (54%)	6 (46%)	1.21 (0.36–3.95), *p* = 0.745	0.17 (0.01–1.37), *p* = 0.132
*Oxygen therapy*	77 (77%)	41 (53%)	36 (47%)	2.49 (0.92–7.51), *p* = 0.084	1.28 (0.40–4.43), *p* = 0.685
*Alveolar recruitment*	19 (19%)	2 (11%)	17 (89%)	19.04 (4.97–125.99), *p* < 0.001	22.34 (3.56–224.91), *p* = 0.002
*Awake prone*	9 (9%)	2 (22%)	7 (78%)	5.60 (1.27–39.05), *p* = 0.038	13.41 (1.62–228.22), *p* = 0.032
*Length of stay, days*	6.2 (6.8)	6.2 (6.8)	14.5 (21.7)	1.07 (1.02–1.13), *p* = 0.008	1.03 (0.97–1.12), *p* = 0.329
*APACHE II, score*	30.2 (4.8)	30.3 (4.8)	30.2 (4.9)	1.00 (0.91–1.08), *p* = 0.916	1.04 (0.97–1.12), *p* = 0.325
**Model 2, Mobility**				**AIC = 138.4, C-statistic = 0.694**	
*Restricted mobility*	61 (61%)	34 (56%)	27 (44%)	1.27 (0.56–2.92), *p* = 0.567	0.83 (0.27–2.48), *p* = 0.739
*Kinesiotherapy*, *passive*	51 (51%)	27 (53%)	24 (47%)	1.53 (0.69–3.44), *p* = 0.297	1.81 (0.42–8.45), *p* = 0.432
*Kinesiotherapy*, *active*	68 (68%)	38 (56%)	30 (44%)	1.32 (0.56–3.17), *p* = 0.532	1.17 (0.20–6.17), *p* = 0.852
*Standing*	60 (60%)	32 (53%)	28 (47%)	1.62 (0.72–3.77), *p* = 0.248	1.88 (0.30–14.59), *p* = 0.514
*Walking*	47 (47%)	25 (53%)	22 (42%)	1.45 (0.65–3.25), *p* = 0.360	1.45 (0.40–5.58), *p* = 0.577
*Length of stay*, *days*	6.2 (6.8)	6.2 (6.8)	14.5 (21.7)	1.07 (1.02–1.13), *p* = 0.008	1.07 (1.01–1.14), *p* = 0.029
*APACHE II*, *score*	30.2 (4.8)	30.3 (4.8)	30.2 (4.9)	1.00 (0.91–1.08), *p* = 0.916	0.98 (0.89–1.08), *p* = 0.748

Mean (SD), Absolute frequency (%) or odds ratio (OR) with 95% confidence interval (95% CI). AIC: Akaike information criterion. Bold formatting represents overall model results. Italic formatting represents individual variables within a model.

**Table 3 medicina-58-00823-t003:** Association (odds ratio and 95% confidence intervals) of in-ICU outcome (death vs. discharge) and admission functional measurements with exposure to the physiotherapy interventions (yes vs. no) estimated using univariable and multivariable logistic regression models (*n* = 100).

Exposures	All Participants	Outcomes		OR (95% CI)	
		ICU Discharge	In-ICU Death	Univariable	Multivariable
**Model 1, Ventilatory support**				**AIC = 95.6, C-statistic = 0.920**	
*COVID-19+ test result*	42 (42%)	20 (48%)	22 (52%)	3.15 (1.37–7.47) *p* = 0.008	5.51 (1.25–28.43), *p* = 0.029
*Invasive mechanical ventilation*	40 (40%)	10 (25%)	30 (75%)	22.71 (8.28–70.76), *p* < 0.001	14.81 (2.97–93.70), *p* = 0.002
*Invasive mechanical ventilation*, *in prone*	13 (13%)	4 (31%)	9 (69%)	4.74 (1.42–18.74), *p* = 0.016	0.75 (0.07–9.45), *p* = 0.813
*Noninvasive mechanical ventilation*	13 (13%)	6 (46%)	7 (54%)	2.22 (0.68–7.46), *p* = 0.185	0.54 (0.07–4.01), *p* = 0.535
*Oxygen therapy*	77 (77%)	42 (55%)	35 (45%)	8.75 (2.34–57.11), *p* = 0.005	2.61 (0.43–24.87), *p* = 0.334
*Alveolar recruitment*	19 (19%)	4 (21%)	15 (79%)	10.06 (3.25–38.35), *p* < 0.001	1.68 (0.21–17.57), *p* = 0.636
*Awake prone*	9 (9%)	4 (44%)	5 (56%)	2.30 (0.57–9.89), *p* = 0.237	1.88 (0.17–17.51), *p* = 0.584
*Admission MRC*, *score*	45.7 (9.6)	47.1 (9.2)	43.2 (10.0)	0.96 (0.91–1.00), *p* = 0.055	0.96 (0.89–1.03), *p* = 0.316
*Admission IMS*, *score*	4.6 (3.9)	5.7 (3.8)	2.8 (3.5)	0.81 (0.71–0.91), *p* = 0.001	0.79 (0.63–0.96), *p* = 0.023
*Length of stay*, *days*	100 (100%)	7.0 (9.1)	14.2 (22.0)	1.04 (1.01–1.09), *p* = 0.048	0.99 (0.95–1.03), *p* = 0.629
*APACHE II*, *score*	30.2 (4.8)	29.8 (4.2)	31.0 (5.7)	1.05 (0.97–1.15), *p* = 0.235	1.00 (0.88–1.15), *p* = 0.950
**Model 2, Mobility**				AIC = 81.6, C-statistic = 0.947	
*COVID-19+ test result*	42 (42%)	20 (48%)	22 (52%)	3.15 (1.37–7.47), *p* = 0.008	15.44 (2.80–140.82), *p* = 0.005
*Restricted mobility*	61 (61%)	26 (43%)	35 (57%)	24.90 (6.77–161.94), *p* < 0.001	10.49 (1.74–95.85), *p* = 0.017
*Kinesiotherapy*, *passive*	51 (51%)	17 (33%)	34 (67%)	30.67 (9.49–139.52), *p* < 0.001	17.66 (2.32–326.56), *p* = 0.017
*Kinesiotherapy*, *active*	68 (68%)	53 (78%)	15 (22%)	0.13 (0.05–0.32), *p* < 0.001	0.92 (0.11–8.30), *p* = 0.937
*Standing*	60 (60%)	49 (82%)	11 (18%)	0.12 (0.05–0.30), *p* < 0.001	0.68 (0.04–15.04), *p* = 0.791
*Walking*	47 (47%)	41 (87%)	6 (13%)	0.10 (0.03–0.27), *p* < 0.001	0.07 (0.00–1.19), *p* = 0.104
*Admission MRC*, *score*	45.7 (9.6)	47.1 (9.2)	43.2 (10.0)	0.96 (0.91–1.00), *p* = 0.055	0.99 (0.91–1.06), *p* = 0.713
*Admission IMS*, *score*	4.6 (3.9)	5.7 (3.8)	2.8 (3.5)	0.81 (0.71–0.91), *p* = 0.001	1.09 (0.83–1.51), *p* = 0.567
*Length of stay*, *days*	100 (100%)	7.0 (9.1)	14.2 (22.0)	1.04 (1.01–1.09), *p* = 0.048	0.97 (0.92–1.01), *p* = 0.192
*APACHE II*, *score*	30.2 (4.8)	29.8 (4.2)	31.0 (5.7)	1.05 (0.97–1.15), *p* = 0.235	1.08 (0.95–1.26), *p* = 0.283

Mean (SD), absolute frequency (%) or odds ratio (OR) with 95% confidence interval (95% CI). AIC: Akaike information criterion. MRC: Medical Research Council. IMS: Intensive care unit Mobility Scale. Bold formatting represents overall model results. Italic formatting represents individual variables within a model.

## Data Availability

Not applicable.

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
