# Peer review of "Association of Functional Characteristics and Physiotherapy with COVID-19 Mortality in Intensive Care Unit in Inpatients with Cardiovascular Diseases"

_medicina, 2022, doi:10.3390/medicina58060823_

Round 1

Reviewer 1 Report

it's a good manuscript about an interesting argument

Author Response

  1. It's a good manuscript about an interesting argument

Thank you for your positive assessment of our manuscript. We hope our argument to be of assistance for the investigated population.

Reviewer 2 Report

The authors of the present article entitled ‘‘Association of functional characteristics and physiotherapy with COVID-19 mortality in intensive care unit in inpatients with cardiovascular diseases’’ aimed to investigate whether functional status on admission and implementation of physiotherapeutic interventions are associated with in-ICU mortality rate in a retrospective cohort of ICU patients with concomitant cardiovascular diseases, divided into 2 groups, namely with versus without COVID-19. The main strength of this study is the fact that the authors try to emphasize the undoubtedly beneficial effects of systematic implementation of physiotherapy in ICU patients with cardiovascular disease, even more so in COVID-19 patients hospitalized in the ICU. However, there are several limitations concerning the study design and the methods utilized. As highlighted by the authors, results are based on retrospective data concerning COVID-19 and non-COVID-19 patients with underlying cardiovascular diseases hospitalized in an ICU environment. Admission criteria to the ICU are rather vague and moreover no specific diagnoses for non-COVID-19 patients that necessitated ICU treatment are provided, while the reader should make the assumption that COVID-19 patients required ICU hospitalization for presumed degrees of respiratory failure. Furthermore, a well-constructed protocol of physiotherapeutic interventions that would be homogeneously and constantly applied to the studied ICU patients is lacking, thus excluding one from drawing safe conclusions. In this context, the results should be interpreted with caution, since they represent an analysis in a non-homogeneous sample of ICU patients with a wide range of disease severity and unspecified diagnoses. Below, several detailed remarks are presented to the authors that need to be addressed.

MAJOR ISSUES

  1. ICU admission criteria in the study are rather ambiguous and should be better defined.
  2. Based on the admission data reported for patients with and without COVID-19, their levels of PaO2 (100.2 and 89.4) and O2 saturation (95.1 and 93.9) do not warrant ICU admission. What was the criterion for admitting these patients to the ICU?
  3. For the ICU patients without COVID-19, what were the underlying diagnoses that required hospitalization in the ICU?
  4. In the Materials and Methods section, some parts are rather vague and need additional clarification.
  5. The fact that ICU admission was defined as an admission to the hospital’s ICU for >12 hours raises some serious questions. By using this criterion, the severity of the condition of the patients admitted to the ICU may vary widely, since it is known that COVID-19 patients with ARDS require prolonged hospitalization, while at the same time a non-COVID-19 patient admitted to the ICU with a diagnosis of acute pulmonary oedema might as well be easily discharged in less than 24 hours if clinically stable. A patient who has been admitted to the ICU and has been discharged in 1-2 days or even more so in less than 24 hours cannot be fairly compared to a patient who requires weeks or months to be discharged.
  6. On the same lines, by defining exposure to each routine physiotherapy intervention as using a given therapeutic resource at any time during the total length of stay in the ICU and registering it as a dichotomous variable (yes or no), certain issues arise. The various physiotherapy interventions should have been applied routinely in an organized manner and at a certain frequency per day for each individual patient. Otherwise, any different approach precludes any further safe comparison between the 2 groups of patients with and without COVID-19 on equal grounds and can lead to questionable conclusions.
  7. It is important to describe the physiotherapeutic interventions in detail, since they constitute a crucial part of the manuscript’s scope.
  8. Statistical analysis should be further refined in order to produce more robust results. It does not take any confounding factors into consideration, such as length of ICU stay. Results are presented without making any adjustments for possible confounders.
  9. In my opinion, the results of any comparison between COVID-19 and non-COVID-19 patients regarding ventilatory support, the exposure to physiotherapeutic interventions and observed ICU mortality are rather arbitrary, since they are confounded by the fact that COVID-19 patients had a longer ICU stay, the disease severity is unknown (no ICU scoring system, such as APACHE or SAPS, is provided), admission criteria are lacking and physiotherapeutic strategies may have not been equally applied in terms of homogeneity and systematic implementation. Thus, the results should be interpreted with caution, since they could be misleading and confusing.
  10. Since the COVID-19 pandemic outbreak, a number of variants have emerged. It would be beneficial to readers to accurately define the SARS-CoV-2 variant that was prevalent during the studied time period.

MINOR ISSUES
1. More attention should be paid on the reported data. There seems to be a discrepancy between the number of patients with COVID-19 (58) mentioned in the abstract (Line 23) and the corresponding number (42) eported in the main text (Tables 1 and 3). Another discrepancy has also been noted in the reported data concerning overweight patients between the value (39) reported in the text (Line 156) and the value (40) depicted in Table 1.
2. When using abbreviations and acronyms, they should first be presented in the expanded form and abbreviated thereafter. This should be done separately for the abstract and for the main text. e.g. Abstract: IMS (Line 26), CVD (Line 30). Text: COVID-19 (Line 51), PEEP (Line 112), SD (Line 139), OR (Line 145), CI (Line 145), BMI (Line 156), PCO2 (Line 164), PaO2 (Table 1).

  1. There are some expression issues and certain parts of the manuscript should undergo English editing.
    4. Please revise the sentence in Lines 62-64 ‘‘Nonetheless, whether admission functional characteristics and physiotherapy interventions can affect, and the risk of in-ICU death remains to be determined in this population.’’
    5. Please revise the sentence in Lines 113-114 ‘‘whereas that they presented clinical stability for such’’.
    6. Please revise the phrase in Line 116 ‘‘in the frontal decubitus position’’.
    7. For purposes of better clarity, please revise the phrase ‘‘manifestations of symptoms’’ in Line 50 in order to delineate that you are referring to more severe manifestation of symptoms.
    8. In order to achieve better writing flow, please revise the punctuation marks in the very long sentence starting from Line 106 and ending in Line 116. The semicolons (;) in Lines 110, 114 and 115 should be replaced with periods (.)
    9. Please correct the phrase ‘‘will be performed’’ in Line 134 to ‘‘was performed’’.
    10. The results for MRC and IMS scores presented in parentheses in Lines 161 and 162 should be reverted since they refer to COVID19 patients versus non-COVID19 and not the other way around, i.e for MRC scores 48.3 [7.9] vs 43.8 [10.4] instead of 43.8 [10.4] vs 48.3 [7.9]. The same should be done for IMS scores. This will ensure keeping in line with the presentation of the aforementioned and subsequent results presented in the same paragraph.
    11. Please add a closed parenthesis in Line 223 after the word ‘‘walking’’.
    12. Please correct the verb ‘‘was missing’’ in Line 269 to ‘‘were missing’’.
    13. Please correct the phrase in Line 279 ‘‘active rather passive than kinesiotherapy’’
    to ‘‘active rather than passive kinesiotherapy’’.
    14. Reference 2 should be revised to the English version, as follows: Brant LCC, Nascimento BR, Passos VMA, Duncan BB, Bensenõr IJM, Malta DC, Souza MFM, Ishitani LH, França E, Oliveira MS, et al. Variations and particularities in cardiovascular disease mortality in Brazil and Brazilian states in 1990 and 2015: estimates from the Global Burden of Disease. Rev Bras Epidemiol. 2017;20Suppl 01:116-128. doi: 10.1590/1980-5497201700050010.
    15. Reference 18 should be properly cited.
    16. Please provide doi number for reference 22.

Author Response

The authors of the present article entitled ‘‘Association of functional characteristics and physiotherapy with COVID-19 mortality in intensive care unit in inpatients with cardiovascular diseases’’ aimed to investigate whether functional status on admission and implementation of physiotherapeutic interventions are associated with in-ICU mortality rate in a retrospective cohort of ICU patients with concomitant cardiovascular diseases, divided into 2 groups, namely with versus without COVID- 19. The main strength of this study is the fact that the authors try to emphasize the undoubtedly beneficial effects of systematic implementation of physiotherapy in ICU patients with cardiovascular disease, even more so in COVID-19 patients hospitalized in the ICU. However, there are several limitations concerning the study design and the methods utilized. As highlighted by the authors, results are based on retrospective data concerning COVID-19 and non-COVID-19 patients with underlying cardiovascular diseases hospitalized in an ICU environment. Admission criteria to the ICU are rather vague and moreover no specific diagnoses for non-COVID-19 patients that necessitated ICU treatment are provided, while the reader should make the assumption that COVID-19 patients required ICU hospitalization for presumed degrees of respiratory failure. Furthermore, a well-constructed protocol of physiotherapeutic interventions that would be homogeneously and constantly applied to the studied ICU patients is lacking, thus excluding one from drawing safe conclusions. In this context, the results should be interpreted with caution, since they represent an analysis in a non-homogeneous sample of ICU patients with a wide range of disease severity and unspecified diagnoses. Below, several detailed remarks are presented to the authors that need to be addressed.

Thank you for your positive and constructive assessment of our manuscript. We revised the manuscript following your suggestions. Please find below the answers for each comment.

MAJOR ISSUES

  1. ICU admission criteria in the study are rather ambiguous and should be better defined.

The text was revised to: “ICU admission criteria comprised at least of the following conditions: hemodynamic (e.g., symptomatic hypotension, hypertension with target-organ damage; hypovolemia, perfusion impairment of any etiology); neurological (e.g., lowering of consciousness level, intoxication, ischemic or hemorrhagic stroke, convulsive crisis); or respiratory (e.g., hypoxemia, respiratory failure, pneumothorax, pulmonary edema, dyspnea). Other causes for ICU admission included post-operative of high-risk patients, postoperative period of large surgeries, post-operative cardiac surgery, postoperative neurological surgery, postoperative endovascular surgery, need for clinical monitoring, sepsis, septic shock, cardiac arrhythmia, vascular diseases of the heart, or acute renal dysfunction.”.

  1. Based on the admission data reported for patients with and without COVID-19, their levels of PaO2 (100.2 and 89.4) and O2 saturation (95.1 and 93.9) do not warrant ICU admission. What was the criterion for admitting these patients to the ICU?

Thank you for your question. As described in text, clinical data at admission was collected within <24 h of ICU hospitalization, not status before ICU admission. The text in Discussion was revised to: “Clinical data at admission were collected within <24 h of ICU hospitalization and hence may differ from pre-admission status that required hospitalization.”.

  1. For the ICU patients without COVID-19, what were the underlying diagnoses that required hospitalization in the ICU?

Thank you for your question. The following explanation was added to the Results section: “Underlying diagnoses that required hospitalization in the ICU were likely multiple per patient and comprised pulmonary sepsis or pneumonia (43/100); decompensated heart failure (13/100); acute kidney failure (13/100); dyspnoea (9/100); exacerbation of chronic pulmonary obstructive disease (7/100); severe acute respiratory syndrome (6/100); stroke (5/100); pulmonary embolism (5/100); acute respiratory failure, acute myocardial infarction, femur fracture, chest pain, urinary tract infection (each 3/11); abdominal pain, minimally conscious state, electrophysiological study, sepsis (each 2/100); and exploratory laparotomy, lower gastrointestinal bleeding, diabetic ketoacidosis, acute pancreatitis, syncope, soft tissue sepsis, post-surgery shoulder arthroplasty, post-surgery atrial ablation, diverticulitis, acute tachycardia, post-surgery spinal cord stimulator implantation, acute cholecystitis, atrial flutter, atrioventricular block, deep venous thrombosis, post-surgery pericardial drainage, hypoxia, pleural effusion, abdominal sepsis, bronchospasm, atrial fibrillation with rapid ventricular response, and anaemia (each 1/100).”

  1. In the Materials and Methods section, some parts are rather vague and need additional clarification.

Please find below the revisions for your specific comments.

  1. The fact that ICU admission was defined as an admission to the hospital’s ICU for >12 hours raises some serious questions. By using this criterion, the severity of the condition of the patients admitted to the ICU may vary widely, since it is known that COVID-19 patients with ARDS require prolonged hospitalization, while at the same time a non-COVID-19 patient admitted to the ICU with a diagnosis of acute pulmonary oedema might as well be easily discharged in less than 24 hours if clinically stable. A patient who has been admitted to the ICU and has been discharged in 1-2 days or even more so in less than 24 hours cannot be fairly compared to a patient who requires weeks or months to be discharged.

Thank you for your comment. We agree that length of stay could be a confounding factor. We added this variable to the univariable and multivariable models and reported the results. The text in Results and Tables 2 and 3 were revised to match this new updated analysis. The following texts were added:

“Patients with COVID-19 were hospitalized for a longer period (1.07 [1.02-1.13], p = 0.008]) (Table 2). Length of stay remained associated with COVID status in multivariable analysis of mobility (1.07 [1.01-1.14], p = 0.028) but not ventilatory support (1.03 [0.97-1.12], p = 0.329).”

“In-ICU death was associated with a longer length of stay (1.04 [1.01-1.09], p = 0.048]) (Table 3). Statistical evidence of association between length of stay and in-ICU was not observed in multivariable analysis of ventilatory support (0.99 [0.95-1.03], p=0.627) nor mobility (0.97 [0.92-1.01], p = 0.165).”

  1. On the same lines, by defining exposure to each routine physiotherapy intervention as using a given therapeutic resource at any time during the total length of stay in the ICU and registering it as a dichotomous variable (yes or no), certain issues arise. The various physiotherapy interventions should have been applied routinely in an organized manner and at a certain frequency per day for each individual patient. Otherwise, any different approach precludes any further safe comparison between the 2 groups of patients with and without COVID-19 on equal grounds and can lead to questionable conclusions.

Thank you for your comment. We apologize for this omission in the original version. The following description was provided in Methods: “First, a multidisciplinary rehabilitation meeting is held daily to link the plans between the professionals of each patient. In sequence, an initial bedside assessment of vital signs, pain intensity, ventilatory pattern, level of consciousness, and dosage of drug infusion in use is conducted to decide upon the removal of patients from the bed and/or which physiotherapy intervention to perform. In general, physiotherapy is performed 2 to 3 times daily (one service at each staff shift), involving respiratory and mobility interventions. Ventilatory support adjustments were made according to clinical status, laboratory tests; asynchronous adjustments or parameter changes for weaning were personalized.”. Also, the text in Discussion was revised to: “Also, physiotherapy interventions were delivered as per the rehabilitation team’s clinical decision-making process. Whereas such lack of control about the experimental factors may have influenced the delivered interventions in each group, such pragmatic approach most likely represents ICU routines in Brazil since they are based on national guidelines.”

  1. It is important to describe the physiotherapeutic interventions in detail, since they constitute a crucial part of the manuscript’s scope.

Thank you for your suggestion. The text in Methods was revised to: “All patients were exposed to physiotherapy interventions based on the hospital standards of usual care developed according to international and national recommendations [19–28], which are briefly described here. Full descriptions of the hospital standards for usual care are available in Portuguese-Brazil upon request to the authors.”. The complete protocols are submitted as additional material for review only (authorization for publication not granted by the institution).

  1. Statistical analysis should be further refined in order to produce more robust results. It does not take any confounding factors into consideration, such as length of ICU stay. Results are presented without making any adjustments for possible confounders.

Thank you for your suggestion. We added length of stay as an additional confounder as per your comment #5.

  1. In my opinion, the results of any comparison between COVID-19 and non-COVID- 19 patients regarding ventilatory support, the exposure to physiotherapeutic interventions and observed ICU mortality are rather arbitrary, since they are confounded by the fact that COVID-19 patients had a longer ICU stay, the disease severity is unknown (no ICU scoring system, such as APACHE or SAPS, is provided), admission criteria are lacking and physiotherapeutic strategies may have not been equally applied in terms of homogeneity and systematic implementation. Thus, the results should be interpreted with caution, since they could be misleading and confusing.

Thank you for your suggestion. We revised the text considering the clarification of ICU admission criteria, inclusion of length of ICU stay as a confounder in all models, and the description of physiotherapy interventions. We also expanded the discussed some of these issues as limitations for a comprehensive generalization of our findings.

  1. Since the COVID-19 pandemic outbreak, a number of variants have emerged. It would be beneficial to readers to accurately define the SARS-CoV-2 variant that was prevalent during the studied time period.

Thank you for your suggestion. The text was revised to: The current sample from a single centre during the first ‘wave’ (February to November 2020) of cases in Brazil [40] when variants of concern P.1 and B.1.1.7 were the most prevalent ones [41] and may not reflect the nationwide healthcare system and then requires further investigation.

MINOR ISSUES

  1. More attention should be paid on the reported data. There seems to be a discrepancy between the number of patients with COVID-19 (58) mentioned in the abstract (Line 23) and the corresponding number (42) reported in the main text (Tables 1 and 3). Another discrepancy has also been noted in the reported data concerning overweight patients between the value (39) reported in the text (Line 156) and the value (40) depicted in Table 1.

Thank you for pointing out these blunders. The text in Abstract was revised to: “Of total, 42 (42%) patients tested positive for COVID-19.” and “Overall, the sample was composed of older adults (75 [16] years), balanced between sexes (female/male 49/51), and with most participants (40/100) classified as overweight (BMI = 27.1 [5.1] kg/m2).”.

  1. When using abbreviations and acronyms, they should first be presented in the expanded form and abbreviated thereafter. This should be done separately for the abstract and for the main text. e.g. Abstract: IMS (Line 26), CVD (Line 30). Text: COVID-19 (Line 51), PEEP (Line 112), SD (Line 139), OR (Line 145), CI (Line 145), BMI (Line 156), PCO2 (Line 164), PaO2 (Table 1).

Thank you for this suggestion. The text in was revised to (line numbers were updated for the revised version):

“In-ICU death was associated with lower admission ICU Mobility Scale score…”

“In-ICU mortality is higher for inpatients with cardiovascular diseases …”

“…hospitalized with new coronavirus disease (COVID-19) [6–9].”

“…gradual increase of positive end-expiratory pressure (PEEP)…”

“Descriptive summaries were reported as mean (standard deviation [SD])”

“…(odds ratio [OR] with 95% confidence interval [95%CI])…”

“…(body mass index = 27.1 [5.1] kg/m2)…” (PS: abbreviation deleted from here and Table 1)

“…lower partial pressure of carbon dioxide (PCO2)…”

Table 1 (line 199): “PaO2: partial pressure of oxygen”

  1. There are some expression issues and certain parts of the manuscript should undergo English editing.

Thank you for your suggestion. Please find below the changes as per you suggestion.

  1. Please revise the sentence in Lines 62-64 ‘‘Nonetheless, whether admission functional characteristics and physiotherapy interventions can affect, and the risk of in-ICU death remains to be determined in this population.’’

The text was revised to: “Nonetheless, whether functional characteristics at ICU admission and physiotherapy interventions can affect the risk of in-ICU death remains unclear in this population.”

  1. Please revise the sentence in Lines 113-114 ‘‘whereas that they presented clinical stability for such’’.

The text was revised to: “provided they were clinically stable”.

  1. Please revise the phrase in Line 116 ‘‘in the frontal decubitus position’’.

The text was deleted as considered redundant as defined in text.

  1. For purposes of better clarity, please revise the phrase ‘‘manifestations of symptoms’’ in Line 50 in order to delineate that you are referring to more severe manifestation of symptoms.

The text was revised to: “as well as manifestations of more severe symptoms”.

  1. In order to achieve better writing flow, please revise the punctuation marks in the very long sentence starting from Line 106 and ending in Line 116. The semicolons (;) in Lines 110, 114 and 115 should be replaced with periods (.)

The text was revised to: “Ventilatory support was characterized by the use of non-invasive mechanical ventilation, through an orofacial or facial interface connected to the mechanical ventilator in one or two pressure levels ventilation modes; or invasive ventilatory support (orotracheal or tracheal prosthesis in controlled ventilatory modes, controlled assistance and/or spontaneous). Patients diagnosed with acute respiratory distress syndrome used protective strategy ventilatory parameters, which may require alveolar recruitment through the prone position or recruitment through the gradual increase of positive end-expiratory pressure (PEEP) up to 35 cmH2O and subsequent titration of ideal PEEP, provided they were clinically stable. When they needed oxygen therapy, it was performed using a low-flow system (nasal catheter, face mask with reservoir, tracheostomy mask). Spontaneous prone was also used for at least 1h. In the supine position [29], the head was elevated between 30° and 45°. In the prone position, the head was elevated between 10° and 20°.”

  1. Please correct the phrase ‘‘will be performed’’ in Line 134 to ‘‘was performed’’.

The text was revised to: “Statistical analysis was performed…”

  1. The results for MRC and IMS scores presented in parentheses in Lines 161 and 162 should be reverted since they refer to COVID19 patients versus non-COVID19 and not the other way around, i.e for MRC scores 48.3 [7.9] vs 43.8 [10.4] instead of 43.8 [10.4] vs 48.3 [7.9]. The same should be done for IMS scores. This will ensure keeping in line with the presentation of the aforementioned and subsequent results presented in the same paragraph.

The text was revised to: “Between-groups comparisons show that patients with COVID-19+ showed higher admission MRC scores (48.3 [7.9] vs. 43.8 [10.4], p = 0.021) and IMS scores (5.5 [4.0] vs. 3.9 [3.8], p = 0.039). They were also younger (68 [16] vs. 80 [13] years, p < 0.001) and presented at admission with lower leukocytes count (9565 [5473] vs. 13456 [6444] count/mcL, p = 0.002), lower partial pressure of carbon dioxide (PCO2) (31.7 [6.5] vs. 37.7 [8.5], p < 0.001), and lower bicarbonate (21.9 [4.5] vs. 23.8 [4.9] mEq/L, p = 0.044). Sedation at admission was more frequent in COVID-19+ group (52.4% vs. 39.0%, p = 0.020).”

  1. Please add a closed parenthesis in Line 223 after the word ‘‘walking’’.

The text was revised to: “…active mobilizations (kinesiotherapy, standing, or walking)…”.

  1. Please correct the verb ‘‘was missing’’ in Line 269 to ‘‘were missing’’.

The text was revised to: “…there were missing data…”.

  1. Please correct the phrase in Line 279 ‘‘active rather passive than kinesiotherapy’’ to ‘‘active rather than passive kinesiotherapy’’.

The text was revised to: “…active rather than passive kinesiotherapy…”.

  1. Reference 2 should be revised to the English version, as follows: Brant LCC, Nascimento BR, Passos VMA, Duncan BB, Bensenõr IJM, Malta DC, Souza MFM, Ishitani LH, França E, Oliveira MS, et al. Variations and particularities in cardiovascular disease mortality in Brazil and Brazilian states in 1990 and 2015: estimates from the Global Burden of Disease. Rev Bras Epidemiol. 2017;20Suppl 01:116-128. doi: 10.1590/1980-5497201700050010.

The text was revised accordingly.

  1. Reference 18 should be properly cited.

The text was revised to:  18. Kawaguchi, Y.M.F.; Nawa, R.K.; Figueiredo, T.B.; Martins, L.; Pires-Neto, R.C. Perme Intensive Care Unit Mobility Score and ICU Mobility Scale: Translation into Portuguese and Cross-Cultural Adaptation for Use in Brazil. J. Bras. Pneumol. 2016, 42, 429–434, doi:10.1590/s1806-37562015000000301.

  1. Please provide doi number for reference 22.

The 22 was updated as #32 but has no DOI: “32. Teixeira, J.J.M.; Bastos, G.C.F.C.; Souza, A.C.L. de Profile of Hospitalization of the Elderly. Rev. da Soc. Bras. Clínica Médica 2017, 15, 15–20”.

Reviewer 3 Report

Good written presentation of the study conducted by the authors. I would note only one point: One could claim that in general (=in all ICU patients) restricted mobility or passive kinesiotherapy is associated with in-ICU death;thus, explanation for the difference between the 2 groups should be explained either by the clinical characteristics of the groups (more severe ill patients means restricted mobility or passive kinesiotherapy) or by other way. Though the authors clearly note the clinical status differences between the groups; it would be good to add information that could also affect the result: e.g. Did they had any staffing problems, or physiotherapy programming due to, for exampling restriction due to PPE measures,etc.., that could possible also affect the results? In case of COVID ICU care, assumption of sufficient physiotherapy access may not enough;thus, a clear note about, should be added.

Author Response

  1. Good written presentation of the study conducted by the authors.

Thank you for your positive assessment of our manuscript. Please find below our responses to your specific comments.

  1. I would note only one point: One could claim that in general (=in all ICU patients) restricted mobility or passive kinesiotherapy is associated with in-ICU death; thus, explanation for the difference between the 2 groups should be explained either by the clinical characteristics of the groups (more severe ill patients means restricted mobility or passive kinesiotherapy) or by other way. Though the authors clearly note the clinical status differences between the groups; it would be good to add information that could also affect the result: e.g. Did they had any staffing problems, or physiotherapy programming due to, for exampling restriction due to PPE measures, etc.., that could possible also affect the results? In case of COVID ICU care, assumption of sufficient physiotherapy access may not enough; thus, a clear note about, should be added.

Thank you for this interesting comment. The following information was added to the Discussion section of the revised manuscript: “Finally, we did not explore factors others than clinical characteristics of the groups, ventilatory support and exposure to physiotherapy interventions that might explain the present findings. Given global reports [42] of shortage of health professionals and lack of personal protective equipment it is important to emphasize that the physiotherapists were staffed on a 1:10 ratio for ICU beds and had unrestricted access to personal protective equipment during this study period.”

Reviewer 4 Report

The manuscript talk about the association between physiotherapy interventions and COVID-19 in ICU patients, reporting how important is movility in older adults with cardiovascular diseases (CVD) and COVID-19. The reviewer has some comments:

In the abstract sections is not explain what IMS is.

In the introductions, authors should add the information about the % of CVD deaths (additionally to the number 267,635).

In the results section, could the authors explain why the IC is so wide (Table 3, restricted mobility and kineshioterhraphy? Also, invassive mechanical ventilation in prone and noinvasive mechanical ventilation after adjusting by confounding variables, the OR is protective, although is not significant.

In addition, after adjusting for confounding variables, only passive kinesiotherapy is significant. Why the conclusion say " Patients with COVID-19 who further perform standing and walking activities appear to experience a higher survival effect." The conclusion must be modified and be more in agreement with the results obtained.

Author Response

  1. The manuscript talk about the association between physiotherapy interventions and COVID-19 in ICU patients, reporting how important is movility in older adults with cardiovascular diseases (CVD) and COVID-19. The reviewer has some comments:

Thank you for your positive assessment of our manuscript. Please find below our responses to your specific comments.

  1. In the abstract sections is not explain what IMS is.

Thank you for this comment. The text in was revised to: “In-ICU death was associated with lower admission ICU Mobility Scale score…”

  1. In the introductions, authors should add the information about the % of CVD deaths (additionally to the number 267,635).

The text was revised to: “In Brazil, CVD were responsible for 267,635 (29.5% of all causes) deaths in 1990 and 424,058 (31.2% of all causes) deaths in 2015 [2].”

  1. In the results section, could the authors explain why the IC is so wide (Table 3, restricted mobility and kineshioterhraphy? Also, invassive mechanical ventilation in prone and noinvasive mechanical ventilation after adjusting by confounding variables, the OR is protective, although is not significant.

Thank you for your comment. The following explanation was added to the Discussion section: “Some CI retuned wide ranges that suggest a large uncertainty about the effect (likely due to small cell counts for some predictors), and that further information is needed. Nonetheless, assessment of goodness-of-fit of the models (Akaike information criterion, and C-statistics) suggest acceptable model validity.”

  1. In addition, after adjusting for confounding variables, only passive kinesiotherapy is significant. Why the conclusion say " Patients with COVID-19 who further perform standing and walking activities appear to experience a higher survival effect." The conclusion must be modified and be more in agreement with the results obtained.

Thank you for your comment. As per Reviewer #2 suggestion, we added length of ICU stay as an additional confounder. Such inclusion only changed the finding regarding passive kinesiotherapy in multivariable analysis. We therefore did not make additional changes to the conclusions on this matter.

Round 2

Reviewer 2 Report

I appreciate the fact that the authors tried to provide detailed information about several ambiguous methods utilized in the particular study. However, certain critical points still remain rather vague to me.

1. The fact that there was such a wide variety of ICU underlying diagnoses renders the studied sample of patients extremely heterogeneous, which precludes any comparison between COVID-19 and non-COVID-19 patients. Comparison of ICU patients with such a diversity of pathologies and with varying degrees of clinical severity, just by dividing them in COVID-19 and non-COVID-19 patients, is unequal and most probably leads to arbitrary conclusions. Therefore, comparing functional characteristics and physiotherapeutic interventions in such a divergent population of ICU patients might as well lead to erroneous results.  In order to produce robust results and conclusions, the study design should have included only COVID-19 patients divided into 2 groups, with or without CVD. In this way, any resultant comparison would have been made on equal grounds and would have led to safe conclusions.

2. The fact that the investigators used clinical and laboratory parameters (vital signs, arterial blood gases, laboratory exams) collected within 24 hours prior to ICU admission, and not on ICU admission, does not appropriately reflect the clinical severity of the admitted patients and produces further confusion. 

3. Along the same lines, I cannot comprehend how more than half of the patients (52.4%) were already receiving vasoactive drugs and sedation <24 h prior to the ICU admission, yet they maintained excellent mean vital signs (blood pressure, heart rate, respiratory  rate, O2 saturation) and outstanding arterial gases.

4. Another issue is how the variables included in the multivariate model were selected. The authors fail to mention their criteria and the number of variables their sample size would allow them to include.

5. Unfortunately, no ICU scoring system, like APACHE or SAPS, is provided, which is very crucial when attempting to find an association between functional characteristics and physiotherapy with mortality. As such, it should have also been included in the multivariate analysis.  

Author Response

I appreciate the fact that the authors tried to provide detailed information about several ambiguous methods utilized in the particular study. However, certain critical points still remain rather vague to me.

Thank you for your careful assessment of our reporting. Please find below the revisions based on your new comments.

  1. The fact that there was such a wide variety of ICU underlying diagnoses renders the studied sample of patients extremely heterogeneous, which precludes any comparison between COVID-19 and non-COVID-19 patients. Comparison of ICU patients with such a diversity of pathologies and with varying degrees of clinical severity, just by dividing them in COVID-19 and non-COVID-19 patients, is unequal and most probably leads to arbitrary conclusions. Therefore, comparing functional characteristics and physiotherapeutic interventions in such a divergent population of ICU patients might as well lead to erroneous results. In order to produce robust results and conclusions, the study design should have included only COVID-19 patients divided into 2 groups, with or without CVD. In this way, any resultant comparison would have been made on equal grounds and would have led to safe conclusions.

Thank you for your comment. We agree that such criteria may introduce heterogeneity between groups, but it also reflects a pragmatic approach that more likely reflects the clinical practice as already discussed. The following text from last revision highlights this aspect: “Whereas such lack of control about experimental factors may have influenced the delivered interventions in each group, such pragmatic approach most likely represents ICU routines in Brazil since they are based on national guidelines.”.

Also, we did not enroll patients based on their admission diagnosis, but based on the presence of underlying CVD as this population is considered at a higher risk of adverse events. We believe that including patients with/without CVD and underlying COVID-19 would raise a different study question than including patients with/without COVID-19 and underlying CVD, which should be further investigated in another trial.

  1. The fact that the investigators used clinical and laboratory parameters (vital signs, arterial blood gases, laboratory exams) collected within 24 hours prior to ICU admission, and not on ICU admission, does not appropriately reflect the clinical severity of the admitted patients and produces further confusion.

Thank you for the opportunity to elaborate on this comment. The complete panel of clinical and laboratory parameters – including RT-PCR – was obtained throughout the first 24 h of admission based on the medial staff request. The following revision was added to the manuscript: “All admission data were collected within <24 h of ICU hospitalization at the discretion of the medical staff and covered the required time for swab analysis.”.

  1. Along the same lines, I cannot comprehend how more than half of the patients (52.4%) were already receiving vasoactive drugs and sedation <24 h prior to the ICU admission, yet they maintained excellent mean vital signs (blood pressure, heart rate, respiratory rate, O2 saturation) and outstanding arterial gases.

Thank you for your comment. As mentioned above, the clinical stability and medication status of the sample can be explained by the 24h window after ICU admission (not <24h prior to the ICU admission as stated).

  1. Another issue is how the variables included in the multivariate model were selected. The authors fail to mention their criteria and the number of variables their sample size would allow them to include.

Thank you for pointing out this omission. Sample size was estimated based on the precision of the overall risk estimates rather than the number of events per variable which is still controversial. The following text was added to the Statistical Analysis section: “Minimum sample size was determined to ensure precision of the estimate of overall risk using logistic regression models [17]. A minimum of 96 participants is required to ensure a margin of error ≤ 0.1 for a true outcome proportion equal to 0.5.”

  1. Unfortunately, no ICU scoring system, like APACHE or SAPS, is provided, which is very crucial when attempting to find an association between functional characteristics and physiotherapy with mortality. As such, it should have also been included in the multivariate analysis.

Thank you for your suggestion. We revised the manuscript to report the APACHE II score. The following text was added to the Results section: “No statistical evidence of difference was observed between groups for severity of disease based on acute physiology and chronic health evaluation(APACHE II: 30.3 [4.8] vs 30.2 [4.9], p=0.917).” We also included APACHE II score in the multivariate models as suggested and the main findings remain unchanged. Please refer to Tables 1, 2 and 3 updated with the baseline and model outcomes revised values.

Reviewer 4 Report

The suggestions have been answered properly.

Author Response

  1. The suggestions have been answered properly.

Thank you for your positive assessment of our manuscript.